# Methotrexate as a corticosteroid-sparing agent in leprosy reactions: A French multicenter retrospective study

**Léa Jaume** [1]*, **Estelle Hau**[1], **Gentiane Monsel**[2], **Antoine Mahé**[3], **Antoine Bertolotti**[4], **Antoine Petit**[1], **Britney Le**[5], **Marie Chauveau**[6], **Elisabeth Duhamel**[7], **Thierry Maisonobe**[8], **Martine Bagot**[1], **Jean-David Bouaziz**[1,5], **Faïza Mougari**[9], **Emmanuelle Cambau**[9], **Marie Jachiet**[1,5]*, **Groupe d'infectiologie en dermatologie et des infections sexuellement transmissibles (GrIDIST)**

1 Dermatologie et vénérologie, Hôpital Saint-Louis, Université Paris Cité, Paris, France, 2 Maladies infectieuses et tropicales, Hôpital Pitié-Salpêtrière, Paris, France, 3 Dermatologie et vénérologie, CH Colmar, Colmar, France, 4 Dermatologie et vénérologie, CHU de la Réunion, Saint-Pierre, France, 5 INSERM Human Immunology, Physiopathology and Immunotherapy U976, Hospital Saint-Louis, Paris, France, 6 Médecine interne, CH Saint Nazaire, Saint-Nazaire, France, 7 Medecine interne CH Saint Brieuc, Saint-Brieuc, France, 8 Neurophysiologie clinique, Pitié-Salpêtrière, Paris, France, 9 Service de Mycobactériologie spécialisée et de référence, CNR des mycobactéries (laboratoire associé), APHP GHU Nord; Université Paris Cité, INSERM IAME UMR 1137 -Paris, France

* Lea.jaume@aphp.fr (LJ); Marie.jachiet@aphp.fr (MJ)

**Data Availability Statement:** The data that support the findings of this study are available on request from the corresponding author and/or the general contact email address: arc.leprosy@aphp.fr The

## Abstract

### Introduction

Leprosy reactions (LRs) are inflammatory responses observed in 30%-50% of people with leprosy. First-line treatment is glucocorticoids (GCs), often administered at high doses with prolonged courses, resulting in high morbi-mortality. Methotrexate (MTX) is an immunomodulating agent used to treat inflammatory diseases and has an excellent safety profile and worldwide availability. In this study, we describe the efficacy, GCs-sparing effect and safety of MTX in LRs.

### Methods

We conducted a retrospective multicentric study in France consisting of leprosy patients receiving MTX for a reversal reaction (RR) and/or erythema nodosum leprosum (ENL) since 2016. The primary endpoint was the rate of good response (GR) defined as the complete disappearance of inflammatory cutaneous or neurological symptoms without recurrence during MTX treatment. The secondary endpoint was the GCs-sparing effect, safety and clinical relapse after MTX discontinuation.

### Results

Our study included 13 patients with LRs (8 men, 5 women): 6 had ENL and 7 had RR. All patients had had at least one previous course of GCs and 2 previous treatment lines before starting MTX. Overall, 8/13 (61.5%) patients had GR, allowing for GCs-sparing and even GCs withdrawal in 6/11 (54.5%). No severe adverse effects were observed. Relapse after

date are not publicly available due to privacy or ethical restrictions.

**Funding:** The authors received no specific funding for this work.

**Competing interests:** The authors have declared that no competing interests exist.

MTX discontinuation was substantial (42%): the median relapse time was 5.5 months (range 3–14) after stopping treatment.

## Conclusion

MTX seems to be an effective alternative treatment in LRs, allowing for GCs-sparing with a good safety profile. Furthermore, early introduction during LRs may lead to a better therapeutic response. However, its efficacy seems to suggest prolonged therapy to prevent recurrence.

## Author summary

Leprosy reactions (LRs) are acute inflammatory episodes that complicate 30% to 50% of *Mycobacterium leprae* infections. Type 1 is known as reversal reaction (RR), and type 2 is known as erythema nodosum leprosum (ENL). The two types have relatively distinct pathogenesis and clinical features; furthermore, they can advance to become severe and progress to irreversible nerve damage and deformities. Beyond leprosy infection, LR can bring about neurological burden, poor quality of life, and risk of further descent into poverty for often young patients. Despite effective standardized treatment of leprosy infection with multidrug therapy (MDT) since 1981, treating LRs prove to be challenging. First-line treatment of LR is based on prolonged systemic glucocorticoids (GCs) therapy. The current challenge in leprosy is now to develop and evaluate new therapeutic alternatives to reduce not only the sequelae and morbidity of LR but also the lasting side effects of GCs. Thalidomide is an effective alternative in moderate to severe ENL, but its use is limited given its numerous adverse effects. Methotrexate (MTX) is an immunomodulating agent used to treat inflammatory diseases and has an excellent safety profile and worldwide availability. Here, we describe the efficacy, GCs-sparing effect, and safety of MTX in LR.

## Introduction

Leprosy is an infectious disease caused by an acid-fast bacillus of the *Mycobacterium leprae* complex, including *M. leprae* and *M. lepromatosis* [1].

Leprosy reactions (LRs) result from the host immune response to *M. leprae* complex infection and may concern 30% to 50% of all leprosy patients [2–4]. There are two types of LRs that can both evolve to become severe and progress to irreversible nerve damage and deformities. Type 1 reaction, also known as reversal reaction (RR), is a sudden restoration of cell-mediated immunity, triggering an abrupt shift of the inflammatory reaction (from T helper 2 (Th2) cells to T helper 1 (Th1) cells) toward the tuberculoid pole; this shift causes greater risk of tissue damage due to compressive granulomatous formation [5]. Clinically, RR is an acute and painful infiltration of pre-existing skin lesions with potentially severe and irreversible nerve damage. Type 2 reaction, also known as erythema nodosum leprosum (ENL), is a cutaneous and systemic vasculitis linked with immune complex deposits. Clinically, it results in painful inflammatory nodules associated with systemic symptoms such as fever, arthritis, iritis, neuritis, and lymphadenitis.

Despite effective standardized treatment for leprosy infection with antibiotic combination (i.e., MDT) since 1981, treating LRs remains challenging. First-line treatment is primarily based on prolonged systemic glucocorticoids (GCs) therapies, associated with major morbidity

and sometimes, even death [6]. The current challenge in leprosy is to develop and evaluate new therapeutic alternatives to reduce the sequelae and morbidity of LR [7].

Second-line drugs such as azathioprine and ciclosporin in RR and thalidomide or high-dose of clofazimine in ENL are considered alternatives in GCs-resistant LR, but their use is limited due to their adverse effects [8], [9], [10], [11], [12], [13]. Therapeutic alternatives such as methotrexate (MTX), pentoxifylline, phosphodiesterase-4 inhibitors and tumor necrosis factor-alpha (TNFα) inhibitors have been reported as second-line treatments in ENL in small-scale studies [14], [15].

MTX is an immunomodulating agent used to treat malignant and inflammatory diseases [16] [17]. Its efficacy has been described in 30 LRs cases in the literature, in 7 observations [18–25]. MTX is a cheap molecule, is widely available in many countries, has a good tolerance profile, and is included in the WHO model list of essential medicines [26].

This is the reason why we performed a retrospective study to describe the efficacy, GCs-sparing effect, and safety of MTX for treating LRs.

## Methods

### Ethic statement

This study was conducted in compliance with the Good Clinical Practice protocol and the Declaration of Helsinki principles. The study was approved by the local ethics committee (CPP Paris île de France 4), who waived the requirement for informed consent.

### Study design and inclusion criteria

We conducted a multicentric retrospective study carried out after a national case call (through the GRIDIST (Infectious and Sexually Transmitted Infections Group of the French Society of Dermatology) and the national multidisciplinary leprosy meeting. All leprosy patients with type 1 or type 2 LR treated at least 3 months with MTX between June 2016 and August 2021 in France were included.

### MTX regimen and monitoring

All patients had a mandatory assessment of full blood cell count, serum transaminase levels, serum creatinine with computation of creatinine clearance, serological test for the hepatitis viruses B and C, serum albumin assay and a chest radiograph before starting MTX.

Regarding dose regimen, MTX was given at a minimal dose of 0.15 up to 0.3 mg/kg/week, as typically administered in inflammatory diseases [16].

Monitoring on MTX was consistent with current recommendations [27] (full blood cell counts, serum transaminase and creatinine assays at least one a month for the first 3 months then every 4–12 weeks). Folate supplementation was given at a minimal dosage of 5 to 10 mg once a week at a distance from MTX dose [28].

### Data collection and definitions

Clinical and biological information was obtained from digital medical records for each patient. Recorded data included date of diagnosis of leprosy, gender, country of origin, type of leprosy according to the Ridley-Jopling classification, anti-bacillary treatments received, type of LR (RR or ENL), date of first episode of LR and characteristics of clinical manifestations including cutaneous and neurological. We also collected the LRs treatment before MTX, ongoing treatment at the time of MTX initiation, timing and route of administration of MTX, relapse on and after MTX therapy, alternative treatments after relapse, and MTX ineffectiveness.

Adverse events (AEs) were retrospectively collected and graded according to the Common Terminology Criteria for Adverse Events (CTCAE) v5.0.

As there is no standard recommendation to evaluate MTX therapeutic response, we have used clinically valuable outcomes to evaluate MTX clinical response for this study. We defined three types of clinical response to MTX. Good response (GR) was defined as achieving complete regression of inflammatory symptoms (skin, neurological and/or systemic) and having no relapse during MTX therapy. Partial response (PR) was defined as not achieving complete resolution of inflammatory symptoms or showing relapse during MTX therapy, requiring another add-on treatment to MTX or resulting in GCs dependence under MTX. Non-response (NR) was described as persistent symptoms and/or relapse during MTX treatment, requiring MTX discontinuation or a switch to a therapeutic alternative.

Data are presented as median and mean (range) for quantitative variables and frequency (percentage) for categorical variables.

## Outcomes

The primary endpoint was the GR rate. Secondary endpoints were the GCs-sparing effect, number of relapse after MTX discontinuation in GR and PR groups and MTX tolerance profile.

## Results

### Patient characteristics (Table 1)

A total of 13 patients (8 men, 5 women) fulfilled inclusion criteria and received MTX. Median age at leprosy diagnosis was 29 years (range 12–63). The LRs types were ENL 46% (n = 6) and RR 54% (n = 7). All patients with ENL (n = 6) had a cutaneous presentation such as painful nodules (n = 6) and/or peripheral oedema (n = 4). Extra-cutaneous symptoms were fever (n = 3), polyarthritis (n = 2), and neuropathy (n = 1).

Patients with RR (n = 7) mostly had cutaneous and neurological manifestations (n = 6) and one had a pure neurologic form (n = 1). Two of which had uveitis and one had polyarthritis.

### Previous treatments received for LR

Regarding leprosy, all patients were treated with MDT daily for a median of 2.3 years (range 1.12–7.5). They all received rifampicine and clofazimine. Clofazimine was continued after completion of MDT to prevent LRs and was not included in the median time of MDT treatment. Twelve (92%) patients received disulone, one of them was switched to ofloxacin due to a dapsone-induced anemia. One patient had a G6PD deficiency and was treated with clarithromycin. Aside from clofazimine, no MDT treatment was given during MTX therapy.

Regarding LRs, the median number of treatment lines received before starting MTX was 2 (range 1–4). All patients had received at least one previous course of GCs. The other previous treatments were thalidomide (n = 9, 69%), pentoxifylline (n = 4, 31%), and colchicine (n = 2, 15%).

### Treatment continued with MTX

At the time of MTX initiation, 11 (85%) patients were taking GCs with a mean dose of 46 mg (0–90), 4 patients (31%) were under thalidomide with a mean dose of 112 mg (50–200) and one had pentoxifylline (8%).

**Table 1. Characteristics of patients with leprosy (n = 13).**

| *Socio-demographic features* | |
|---|---|
| Female / male | 5/8 |
| Age at diagnosis (years), median (range) | 29 (12–63) |
| Type of leprosy reaction:<br>1. ENL<br>2. RR | <br>6 (46%)<br>7 (54%) |
| *Historical treatment of LR* | |
| Median number of systemics before starting MTX, (range) | 2 [1–4] |
| GCs | 100% |
| Thalidomide | 9/13 (69%) |
| Others* | |
| *Treatments continued with MTX:* | |
| GCs | 11 (85%) |
| Mean dose of GCs at MTX initiation (range) | 46 (10–90) |
| Thalidomide | 4 (31%) |
| Mean dose of thalidomide at MTX initiation | 112.5 (50–200) |
| Pentoxifylline | 1 (9%) |
| *MTX initiation* | |
| Median time between 1st episode of reaction and MTX introduction (months) | 23 (1.6–131) |
| Median dose of MTX (mg/wk) | 20 [12–25] |
| MTX administration mode<br>    Oral<br>    Subcutaneous<br>    Not reported | 4/11 (36%)<br>7/11 (64%)<br>2/13 |
| Leprosy reaction on MTX<br>    RR<br>  v ENL | 3/13 (23%)<br>1/13 (7%)<br>2/13 (15%) |
| *Evolution during MTX therapy* | |
| GCs weaning under MTX | 6/11 (55%) |
| Duration of weaning GCs (months), median (range) | 6 [3–42] |
| Mean dose of GCs (mg) at MTX discontinuation or last follow up in all patients | 4 (0–20) |
| *Evolution after stopping MTX* | |
| Follow-up after stopping MTX (months), median (range) | 17.5 [3–25] |
| Relapse after stopping MTX among good and partial responders | 3/7 (42%) |
| Time to relapse after stopping MTX (months), median (range) | 5.5 [3–14] |

ENL: erythema nodosum leprum, MTX: methotrexate, RR: reverse reaction, mg/kg: milligrams per kilos body weight, GCs: glucocorticoids

* Pentoxifylline 4/13 (31%), colchicine 2/13 (15%)

## MTX efficacy on LR

**MTX was initiated after a median time of 23 months of LR evolution (1.6–131).** MTX was prescribed subcutaneously in 7 (69%) cases, at a median dose of 20 mg/week (range 12–25) corresponding to 0.29 mg/kg body weight (0.18–0.52), for a median time of 14 months (range 5–50).

GR was observed in 8 (62%) patients. They achieved complete regression of inflammatory symptoms (skin, neurological, and systemic) without relapse during MTX treatment.

Two patients (15%), *(patient 7 and patient 13)* had a PR because they did not achieve complete disappearance of symptoms or presented relapse during MTX treatment, requiring another add-on treatment to MTX or resulting in GCs dependence under MTX.

Three patients (23%) were defined as NR due to persistent symptoms and/or relapse during MTX treatment, requiring discontinuation of MTX or a switch to another alternative therapeutic *(patients 8, 10 and 11)*.

In the GR subgroup (n = 8), MTX was introduced with a median of 22 months after LR diagnosis (range 1.5–119 months) *versus* 54 months (17–131 months) in the PR and NR subgroup (n = 5). The administration mode in the GR group was mainly subcutaneous (n = 7/8) *versus* oral (n = 1/8), whereas in the PR and NR subgroup, the administration mode was predominantly oral (n = 4/5, not reported in one case).

The number of previous treatments received did not differ between the groups, with a median number of 2 previous treatments for both groups and a median dose of MTX of 0.25 mg/kg (0.20–0.35) in the GR group *versus* 0.27 mg/kg (0.18–0.32) in the PR and NR group.

### GCs-sparing effect

Six of 11 (54.5%) patients who were under GCs at MTX introduction achieved GCs withdrawal. The median time to complete withdrawal was 6 months (range 3–42). In all patients, the mean dose of GCs at MTX initiation was 46 mg/day (range 0–90), and was reduced to a mean of 4 mg/day (0–20) at MTX discontinuation or on the last follow-up for patients who were still on MTX. The dose of thalidomide was also decreased, from a mean dose of 112 mg (50–200 mg) at MTX initiation to a mean dose of 18 mg (0–50 mg) at MTX discontinuation or on the last follow up.

### MTX discontinuation and relapse

The median time of follow-up after MTX discontinuation was 17.5 months (range 3–25).

Among the 10 patients with GR or PR, 7 patients (70%) completed their treatment and 3 (30%) are still under MTX treatment. MTX was stopped because of side effects despite efficacy in 3 patients, and in 4 patients (3 women, 1 man) who were planning to have a child after a prolonged remission. Among the 7 patients who stopped MTX after good or partial response, 3 (42%) showed relapse after a median of 5.5 months (range 3–14) *(patients 2,4 and 9)*.

Regarding therapeutic switches after relapse in GR and PR: one patient was treated with a new GCs course, one patient with apremilast and one patient with certolizumab pegol.

Regarding TNF alpha inhibitors, 2 patients received certolizumab pegol in the GR group (one following relapse after MTX discontinuation and one as an alternative to MTX in the context of pregnancy). In the NR group, one patient received infliximab in addition to MTX, and one stopped MTX and started apremilast. The long-term outcome was favourable for all of them. No adverse events were reported with those treatments.

### Safety profile and adverse events

A total of 3 (23%) patients had AEs related to MTX. One patient had a grade II cytolysis that developed in a context of hepatitis B virus chronic coinfection; laboratory monitoring showed a complete recovery after stopping MTX. One patient had a grade I gastrointestinal disorder with nausea requiring MTX discontinuation. One patient had a severe grade III SARS-CoV-2 infection. This AE was considered to be partly related to several previous comorbidities such as severe hypertension, chronic renal failure and heart failure. Regarding thalidomide, grade II neuropathy was reported in two patients. Patient characteristics are summarized in **Tables 1 and 2** and **Fig 1**.

**Table 2. Details of clinical evolution per participant.**

| | Type of leprosy | Type of leprosy reaction | Previous treatment for LR | MTX administration | Minimun and maximum dose of MTX: mg/wk (mg/kg body weight) | MTX duration (months) | Time between 1st LR and MTX (months) | MTX efficacy | Side effects | MTX status (months) | Relapse after MTX discontinuation (months) | Treatment required after MTX failure or discontinuation |
|---|---|---|---|---|---|---|---|---|---|---|---|---|
| 1 | BT | RR skin and neurological (facial diplegia) | GCs | SC | 15–20 mg (0.25–0.3) | 14 | 32 | GOOD RESPONSE GCs withdrawal at 9 months of MTX introduction | None | Ongoing [14] | | |
| 2 | LL | RR skin and neurological (sensorimotor neuropathy) | GCs, thalidomide | SC | 20–25 mg (0.3–0.4) | 17 | 23.5 | GOOD RESPONSE | Gastro-intestinal disorder (nausea) | Completed | Neurological RR relapse [8]: worsening of sensory neuropathy at EMG, pain hypoesthesia | GCs Restarted at 30 mg prednisone daily with gradually decreased Currently at 15 mg/day |
| 3 | LL | ENL with severity criteria: edema of the extremities, fever, MAS, episcleritis, polyarthritis | GCs | SC | 15 mg (0.23) | 7 | 21.5 | GOOD RESPONSE GCs withdrawal at 10 months of MTX introduction | Stopped to become pregnant | Completed | No relapse [12] | |
| 4 | LL | ENL skin (cutaneous nodules), fever and uveitis | GCs, pentoxifylline thalidomide, colchicine | SC | 15 mg (0.2) | 17 | 76.5 | GOOD RESPONSE No GCs Thalidomide withdrawal at 17 months of MTX introduction | Stopped to become pregnant Thalidomide neuropathy | Completed | Cutaneous ENL relapse [13] | Certolizumab pegol Currently at 7 months after starting treatment: no relapse to date |
| 5 | BL | ENL with severity criteria (edema of the extremities, cutaneous nodules), fever, MAS and pericarditis | GCs, thalidomide | Oral | 15–20 mg (0.25–0.3) | 33 | 1.6 | GOOD RESPONSE GCs withdrawal at 23 months of MTX introduction Thalidomide withdrawal at 8 months of MTX introduction | Stopped to become pregnant | Completed | No relapse [3] | Certolizumab pegol (MTX stop for desire of pregnancy and high risk of relapse (history of severe ENL with MAS) |
| 6 | LL | RR skin and neurological (sensorimotor neuropathy), uveitis and polyarthritis | GCs Intra-ocular GCs | SC | 15–20 mg (0.2–0.28) | 17 | 21.6 | GOOD RESPONSE No GCs No relapse of uveitis at 12 months of MTX introduction | None | Ongoing [17] | | |

(Continued)

**Table 2.** (Continued)

| | Type of leprosy | Type of leprosy reaction | Previous treatment for LR | MTX administration | Minimun and maximum dose of MTX: mg/wk (mg/kg body weight) | MTX duration (months) | Time between 1st LR and MTX (months) | MTX efficacy | Side effects | MTX status (months) | Relapse after MTX discontinuation (months) | Treatment required after MTX failure or discontinuation |
|---|---|---|---|---|---|---|---|---|---|---|---|---|
| 7 | LL | RR skin and neurological | GCs, thalidomide, pentoxifylline | Oral | 17.5–25 mg (0.23–0.35) | 50 | 58.9 | PARTIAL RESPONSE Adding thalidomide, and thalidomide withdrawal 1.5 year after MTX introduction Pentoxifylline and GCs withdrawal 3.5 years after MTX introduction | Stopped to become pregnant | Completed | No relapse [12] | |
| 8 | LL | ENL: skin (cutaneous nodules) | GCs, colchicine, thalidomide | Oral | 10–20 mg (0.22–0.45) | 9,8 | 22.1 | NON-RESPONSE Chronic ENL with skin involvement with GCs dependence at 12.5 mg prednisone | None | Ongoing [10] | | Infliximab added 9 months after MTX introduction At third infliximab infusion ENL not yet controlled |
| 9 | BL | RR skin and neurological (sensory neuropathy) and recurrent uveitis | GCs, thalidomide, pentoxifylline | SC | 20 mg (0.25) | 5 | 118.9 | GOOD RESPONSE GCs withdrawal at 4 months of MTX introduction | Grade II cytolysis (HBV co-infection) | Completed | Uveitis relapse [4] | Intra-ocular injections of GCs and introduction of apremilast |
| 10 | LL | RR skin and neurological (radial motor mononeuropathy, sensory neuropathy) | GCs, thalidomide | ND | 20–25 mg (0.23–0.3) | 8 | 17.5 | NON-RESPONSE RR with skin involvement with GCs dependence at 15 mg prednisone | None | Completed | Persistent RR skin flare-ups | Thalidomide introduction without efficacy on RR Relayed by apremilast |
| 11 | BL | RR neurological (chronic sensory neuropathy) | GCs | Oral | ND | 11 | 131.1 | NON-RESPONSE No improvement of the sensory neuropathy | None | Completed | Progressive worsening of sensory neuropathy (EMG and clinical) | The relay by pentoxifylline or an anti-TNF-α inhibitors is under discussion |
| 12 | LL | ENL skin (edema of the extremities, cutaneous nodules), polyarthritis and uveitis | GCs, thalidomide, pentoxifylline | SC | 10–20 mg (0.15 0.3) | 6 | 8.7 | GOOD RESPONSE Early discontinuation of all treatments (COVID-19) | COVID-19 | Completed | No relapse [16] | |

(Continued)

Table 2. (Continued)

| # | Type of leprosy | Type of leprosy reaction | Previous treatment for LR | MTX administration | Minimun and maximum dose of MTX: mg/wk (mg/kg body weight) | MTX duration (months) | Time between 1st LR and MTX (months) | MTX efficacy | Side effects | MTX status (months) | Relapse after MTX discontinuation (months) | Treatment required after MTX failure or discontinuation |
|---|---|---|---|---|---|---|---|---|---|---|---|---|
| 13 | LL | ENL skin (edema of the extremities, cutaneous nodules) and neurological (sensorimotor neuropathy) | GCs, thalidomide | Oral | 10–17.5 mg (0.12–0.25) | 28 | 42.1 | PARTIAL RESPONSE 2 relapses: ENL skin and neurological 1 and 3 months after GCs withdrawal GCs dependence at 5 mg prednisone and thalidomide | None Thalidomide neuropathy | Ongoing [28] | | |

LL: lepromatous leprosy, BL: borderline lepromatous leprosy, BT: borderline tuberculoid leprosy, LR: leprosy reaction, ENL: erythema nodosum leprum, RR: reverse reaction, SC: subcutaneous, ND: not defined, MTX: methotrexate, GCs, glucocorticoids; HBV, hepatitis B virus, EMG, electromyography, MAS, macrophagic activation syndrome, TNF, tumor necrosis factor; IV, intravenous

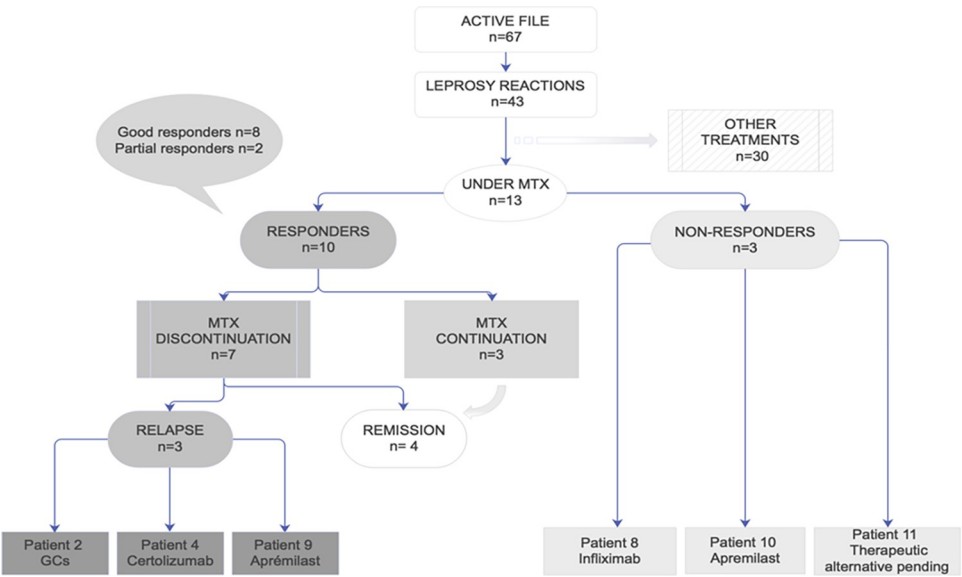

**Fig 1. Flow chart of patients in the study.**

## Discussion

This study is one of the largest cohorts reporting the benefit of MTX for treating LR (both RR and ENL) and allowing for GCs-sparing. In total, 77% of patients (n = 10) were considered responders (GR and PR); 62% showed complete response, defined as a total regression of inflammatory symptoms (cutaneous, neurological and systemic) and no recurrence during treatment. However, the high relapse rate, 42%, after stopping treatment argues for suspensive rather than durable efficacy, which highlights the interest of prolonged MTX treatment to maintain a clinical response, limit relapse upon discontinuation, and avoid iterative courses of GCs. Moreover, MTX could be an option in refractory and/or cortico-dependent LRs and also in earlier stages. TNF-α inhibitors were used in 3 patients, with good safety profile long-term outcome: infliximab was used in addition to MTX for 1 patient and certolizumab pegol was used following relapse after stopping MTX for 2 patients.

We acknowledge a few limitations to our study. Our first limitation is the retrospective design. The second one is the heterogeneity of patient profiles. Certainly, the lack of a standardized protocol in patient management may have reduced subjectivity in the evaluation of treatment response. In the absence of a comparative group, the effect of MTX may have been overestimated, qualifying cases of spontaneous improvement as responses. Our sample size was small although it is the largest cohort in the literature.

Our data were globally concordant with the literature; however some points must be highlighted. Firstly, concerning the MTX course, the dosing and administration regimen differed. Our median MTX dose was twice as high as that in the literature (20 mg/week corresponding to 0.29 mg/ kg body weight *versus* 7.5 mg/week corresponding to 0.14 mg/kg body weight) and was mostly administered subcutaneously, while in contrast, for all the cases in the literature, MTX was given orally. Our therapeutic choice is based on several publications that suggest better efficacy of subcutaneous than oral treatment in inflammatory diseases such as psoriasis and rheumatoid arthritis, with a median dose of 0.3 mg/kg/week usually used in inflammatory diseases [16], [29], [30].

Secondly, the GCs tapering schedule was faster in our patients than in literature case reports (median of 6 *versus* 12 months) with a higher relapse rate under MTX in our cohort (3/13 cases *versus* 2/30 in the literature). Finally, MTX was introduced earlier after a primary episode of LR in the literature cases (median of 11 *versus* 23 months after an initial diagnosis of LR in our cases) but with a similar median of 2 previous treatments. We identified 7 published studies including 30 cases in PubMed. One was a randomized study from Bangladesh comparing the efficacy and safety of MTX plus prednisolone (n = 10 patients) *versus* prednisolone monotherapy (n = 9) during 6 months in ENL. The others were case reports [18–25]. Moreover, the median time on MDT was longer in our patients than in published cases. In our practice, we usually prolong the treatment according to the decrease in the bacillary index, which can be longer than the one to two years recommended by the WHO in order to prevent relapses and LR.

Currently, the first-line treatment of LR is based on prolonged GCs therapy resulting in high cumulative dose, thereby increasing the risk of adverse effects [31] [32]. However, about 40% of individuals with RR do not show clinical improvement; especially with nerve damage, GCs seem to be ineffective [33] [34]. In regards to the management of ENL, the chronic course and frequency of relapses (40% of ENL) are difficult to prevent and often justify prolonged and/or repeated courses of GCs [35].

Azathioprine (AZA) is an immune-suppressant drug used in immune mediated diseases [36]. In AZALEP, a randomised double-blind study of 345 patients in India with RR, adding AZA to GCs did not show improvement of dermatological and neurological outcomes [8].

Ciclosporin has also been tested in a randomised trial in Ethiopia comparing ciclosporin and prednisone efficacy and the tolerance profile in acute RR. The 2 groups did not differ in improved clinical outcomes (skin and nerve), tolerance profile or rate of RR recurrence (85%). Moreover, patients on ciclosporin required additional prednisolone to treat recurrences [9].

Clofazimine is widely used for treating ENL; it has no effect on acute episodes but might be effective in preventing chronic and recurrent ENL [11] [12] [13]. The induced skin pigmentation is sometimes stigmatising and is one of the limiting factors in its long-term use [37]. Thalidomide is another effective therapeutic option in moderate to severe ENL. A retrospective study of 102 patients with ENL in India argues for early initiation of thalidomide to achieve faster and longer remission owing to its efficacy in preventing recurrences [10]. However, Thalidomide would be considered an irrelevant option in leprosy patients with nerve damage due to its neurotoxicity [10,14]. Its use is also limited in young women not on contraception due to its teratogenicity [38] [39].

Other therapeutic alternatives such as pentoxifylline and apremilast (16 cases), an oral phosphodiesterase-4 inhibitor sharing a common molecular structure with thalidomide, seem to avoid recurrence and allow GCs-sparing [15,40–43]

Lastly, some case series concerning the value of biotherapies such as TNF-α inhibitors have recently been published. One review identified 4 published cases of refractory ENL effectively treated with a TNF-α inhibitor (infliximab or etanercept) [14]. All patients previously received thalidomide and prednisone; 2 also received pentoxifylline and one AZA. GCs-sparing was reported in all patients under anti-TNFα, and 3 showed long-term remission after treatment discontinuation. These data are promising and must be confirmed in larger cohorts. Unfortunately, few therapies have demonstrated prolonged efficacy after treatment discontinuation. TNF-α inhibitors seem to have a long-lasting immunological action that can modify the evolutionary trajectory of the disease, but the use of anti-TNF-α is limited by the cost of agents and the availability in developing countries.

In this case series of one of the largest cohorts of patients receiving MTX for LR, we described the potential of MTX as an effective alternative treatment, allowing for GCs-sparing with a good safety profile. Therefore, MTX could be considered as a first-line therapy

combined with GCs to facilitate GCs-sparing and reduce the risk of recurrence and transition to chronic LR. A large randomized controlled trial is underway to assess the efficacy of MTX in conjunction with prednisolone in managing ENL [44].

## Author Contributions

**Conceptualization:** Léa Jaume, Estelle Hau, Antoine Mahé, Thierry Maisonobe, Jean-David Bouaziz, Marie Jachiet.

**Data curation:** Léa Jaume, Marie Jachiet.

**Formal analysis:** Léa Jaume, Marie Jachiet.

**Investigation:** Léa Jaume, Marie Jachiet.

**Methodology:** Léa Jaume, Britney Le, Marie Jachiet.

**Supervision:** Britney Le, Thierry Maisonobe, Jean-David Bouaziz, Faïza Mougari, Emmanuelle Cambau, Marie Jachiet.

**Validation:** Léa Jaume, Estelle Hau, Gentiane Monsel, Antoine Mahé, Antoine Bertolotti, Antoine Petit, Britney Le, Marie Chauveau, Elisabeth Duhamel, Thierry Maisonobe, Martine Bagot, Jean-David Bouaziz, Faïza Mougari, Emmanuelle Cambau, Marie Jachiet.

**Visualization:** Léa Jaume, Gentiane Monsel, Antoine Mahé, Elisabeth Duhamel, Jean-David Bouaziz, Marie Jachiet.

**Writing – original draft:** Léa Jaume, Antoine Petit, Thierry Maisonobe, Martine Bagot, Jean-David Bouaziz, Faïza Mougari, Emmanuelle Cambau, Marie Jachiet.

**Writing – review & editing:** Léa Jaume, Estelle Hau, Gentiane Monsel, Antoine Mahé, Antoine Bertolotti, Antoine Petit, Britney Le, Marie Chauveau, Elisabeth Duhamel, Thierry Maisonobe, Martine Bagot, Jean-David Bouaziz, Faïza Mougari, Emmanuelle Cambau, Marie Jachiet.

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
