## [Decision Letter · Decision Letter 0]

12 Dec 2022

Dear Mme Jaume,

Thank you very much for submitting your manuscript "Methotrexate as a corticosteroid-sparing agent in leprosy reactions: a French multicenter retrospective study" for consideration at PLOS Neglected Tropical Diseases. As with all papers reviewed by the journal, your manuscript was reviewed by members of the editorial board and by several independent reviewers. In light of the reviews (below this email), we would like to invite the resubmission of a significantly-revised version that takes into account the reviewers' comments. 

In addition to the comments submitted by the reviewers please address:

General comments: This manuscript is valuable, but needs rewriting to make it more readable. The use of tense is inconsistent. Please review and provide citations where necessary (mainly in discussion)

Specific comments

Introduction: please

Line 86-101: Please make more succinct. Details can be added in discussion where many of these drugs are discussed again. 

Line 106: There should be a mention that this is a retrospective study of patients treated with LR

Methods:

Line 123: Are these outcomes standard used across the world?If so, please provide citation. Otherwise state that these are outcomes that you have created for this study. 

There is no mention of MTX monitoring or toxicities that were evaluated. 

Line 191: Sub-group analysis is usually done in RCTs, and these are always predefined. suggest removing this sub-heading and simply describing both these groups. 

Line 204: please mention more details about monitoring/lack of monitoring and hepatoxocity seen or not seen. 

Discussion: Need significant editing and rewriting to make more readable. Use of tense is inconsistent

Please highlight that for 3/5 women, the desire to become pregnant necessitated stopping mTX

Line 228-229: should be in the concluding paragraph

Suggest moving lines 233-236 “We identified..” to after line 252

Line 247-250. Speculation without evidence. Please simply mention how your cohort differed from the published literature without any speculation to the reasons such as publication bias. Sug

Lines 253-301: Does not read well currently – please review and reorganize to ensure flow. Start with a discussion of overall outcomes of LR and standard of care with steroids and poor outcomes. Then discuss other options. For each drug, discuss its current use and outcomes from its use. For example lines 253 , 259, 260 can be rewritten into a single sentence. Much of this can be consolidated from the introduction. 

Line 253-254: incorrect tense: GSs do not demonstrate clear benefits (please cite). 

Line 256: “real life experience” – please say “in our experience or our practice” as your experience does not negate others' experience.

Line 259: please cite reference. 

Lines 261: rewrite "Azathioprine acts by inhibiting T-cell immunity. "

Line 265: suspensive is not a commonly used word, and its meaning here is unclear as related to 48 week treatment group. Please clarify and rewrite the entire sentence. 

Line 271-Line 272: should this be in the section on glucocorticoids? Seems out of place between azathioprine and clofazimine. 

Line 273-274: please provide citation. 

Line 278-280: please provide citation. Also rewrite – thalidomide is used with everyone fully knowing its teratogenicity; I doubt it is being prescribed without proper counselling and contraception. This line seems to imply that people are using it in pregnancy and teratogenicity is seen in patients receiving it. 

Line 281-283: could you provide more detail here? How many cases. 

Line 292-295 should be included with summary paragraph about your study at the start of the discussion; currently seems out of place. 

Line 296-301: Please rewrite as a summary paragraph and include “in this case series of one of the largest cohorts of patients receiving MTX for LR, we demonstrate the potential of methotrexate as..." and include line 228

We cannot make any decision about publication until we have seen the revised manuscript and your response to the reviewers' comments. Your revised manuscript is also likely to be sent to reviewers for further evaluation.

Sincerely,

Husain Poonawala

Academic Editor

Joseph Vinetz

Section Editor

In addition to the comments submitted by the reviewers please address:

General comments: This manuscript is valuable, but needs rewriting to make it more readable. The use of tense is inconsistent. Please review and provide citations where necessary (mainly in discussion)

Specific comments

Introduction: please

Line 86-101: Please make more succinct. Details can be added in discussion where many of these drugs are discussed again. 

Line 106: There should be a mention that this is a retrospective study of patients treated with LR

Methods:

Line 123: Are these outcomes standard used across the world?If so, please provide citation. Otherwise state that these are outcomes that you have created for this study. 

There is no mention of MTX monitoring or toxicities that were evaluated. 

Line 191: Sub-group analysis is usually done in RCTs, and these are always predefined. suggest removing this sub-heading and simply describing both these groups. 

Line 204: please mention more details about monitoring/lack of monitoring and hepatoxocity seen or not seen. 

Discussion: Need significant editing and rewriting to make more readable. Use of tense is inconsistent

Please highlight that for 3/5 women, the desire to become pregnant necessitated stopping mTX

Line 228-229: should be in the concluding paragraph

Suggest moving lines 233-236 “We identified..” to after line 252

Line 247-250. Speculation without evidence. Please simply mention how your cohort differed from the published literature without any speculation to the reasons such as publication bias. Sug

Lines 253-301: Does not read well currently – please review and reorganize to ensure flow. Start with a discussion of overall outcomes of LR and standard of care with steroids and poor outcomes. Then discuss other options. For each drug, discuss its current use and outcomes from its use. For example lines 253 , 259, 260 can be rewritten into a single sentence. Much of this can be consolidated from the introduction. 

Line 253-254: incorrect tense: GSs do not demonstrate clear benefits (please cite). 

Line 256: “real life experience” – please say “in our experience or our practice” as your experience does not negate others' experience.

Line 259: please cite reference. 

Lines 261: rewrite "Azathioprine acts by inhibiting T-cell immunity. "

Line 265: suspensive is not a commonly used word, and its meaning here is unclear as related to 48 week treatment group. Please clarify and rewrite the entire sentence. 

Line 271-Line 272: should this be in the section on glucocorticoids? Seems out of place between azathioprine and clofazimine. 

Line 273-274: please provide citation. 

Line 278-280: please provide citation. Also rewrite – thalidomide is used with everyone fully knowing its teratogenicity; I doubt it is being prescribed without proper counselling and contraception. This line seems to imply that people are using it in pregnancy and teratogenicity is seen in patients receiving it. 

Line 281-283: could you provide more detail here? How many cases. 

Line 292-295 should be included with summary paragraph about your study at the start of the discussion; currently seems out of place. 

Line 296-301: Please rewrite as a summary paragraph and include “in this case series of one of the largest cohorts of patients receiving MTX for LR, we demonstrate the potential of methotrexate as..." and include line 228

Reviewer's Responses to Questions

**Key Review Criteria Required for Acceptance?**

**Methods**

-Are the objectives of the study clearly articulated with a clear testable hypothesis stated?

-Is the study design appropriate to address the stated objectives?

-Is the population clearly described and appropriate for the hypothesis being tested?

-Is the sample size sufficient to ensure adequate power to address the hypothesis being tested?

-Were correct statistical analysis used to support conclusions?

-Are there concerns about ethical or regulatory requirements being met?

Reviewer #1: Small sample size

Reviewer #2: Slit skin smear of cases not given

No Histopathology examination done to differentiate Type 1 from 2 reactions as it is unusual to both exist together as was in two of cases

Inclusion criteria not clear

Dose of Methotrexate - how calculated not given

Dose of Corticosteroids nor given

Reviewer #3: (No Response)

**Results**

-Does the analysis presented match the analysis plan?

-Are the results clearly and completely presented?

-Are the figures (Tables, Images) of sufficient quality for clarity?

Reviewer #1: yes results are clear and tables are sufficient quality

Reviewer #2: Outcome parameters seems subjective

How response was attributed to corticosteroids when all the patients were also in corticosteroids and continued for mor e than 6 months in addition to Thalidomide and pentoxifylline in few

Methotrexate is also acting drug in leprosy reactions takes 8-12 weeks to show meaningful clinical response

Reviewer #3: Results need some clarification

**Conclusions**

-Are the conclusions supported by the data presented?

-Are the limitations of analysis clearly described?

-Do the authors discuss how these data can be helpful to advance our understanding of the topic under study?

-Is public health relevance addressed?

Reviewer #1: yes the study mentioned the limitations and addressed the public health

Reviewer #2: Not supported by the study findings

Reviewer #3: Conclusions need to be revised in view of the study limitations

**Editorial and Data Presentation Modifications?**

Reviewer #1: I have mentioned the minor modifications in manuscript pdf, so article can be recommended with minor revision

Reviewer #2: Terminologies and descriptions of some terms is not as used in leprosy in scientific publications. eg Multi-neuritis, Lucio vasculitis in ENL...etc

Reviewer #3: (No Response)

**Summary and General Comments**

Reviewer #1: Nothing specific

Reviewer #2: Not suitable as original article

Reviewer #3: The article is of interest and contributes with valuable information.

PLOS authors have the option to publish the peer review history of their article (what does this mean?). If published, this will include your full peer review and any attached files.

Reviewer #1: Yes: Seema Rani

Reviewer #2: No

Reviewer #3: No
---

## [Decision Letter · Decision Letter 1]

12 Mar 2023

Dear Mme Jaume,

We are pleased to inform you that your manuscript 'Methotrexate as a corticosteroid-sparing agent in leprosy reactions: a French multicenter retrospective study' has been provisionally accepted for publication in PLOS Neglected Tropical Diseases.

Best regards,

Husain Poonawala

Academic Editor

Joseph Vinetz

Section Editor

Reviewer's Responses to Questions

**Key Review Criteria Required for Acceptance?**

**Methods**

-Are the objectives of the study clearly articulated with a clear testable hypothesis stated?

-Is the study design appropriate to address the stated objectives?

-Is the population clearly described and appropriate for the hypothesis being tested?

-Is the sample size sufficient to ensure adequate power to address the hypothesis being tested?

-Were correct statistical analysis used to support conclusions?

-Are there concerns about ethical or regulatory requirements being met?

Reviewer #1: Nothing specific except the small sample size which the author have mentioned in lacunae.

Reviewer #2: Methods - Since MDT was given for median 3.5 years - there is no mention of baseline / annual Slit skin smears (BI) to suggest why it was extended beyond usual recommended schedule.

Spectrum of leprosy is not mentioned.

Which type of MDT given - PB/MB, Details of drugs not given

"Twelve (92%) patients received disulone, one of them was switched to ofloxacin due to a

157 dapsone-induced anemia - is it dapsone?. If Dapsone can not be give, there is no need to replace with any other drug as per WHO recommendations?

"One patient had a G6PD deficiency and was treated with clarithromycin - as there is no such recommendation - please mention reason.

" Aside from clofazimine, no MDT treatment was given during MTX therapy." - please make it clear if all patients were given MTX after completion of MDT (after 2 years?)..what was time of onset of reactions in these patients? what dose of clofazimine was given? why? how long? in all cases? to prevent which type of reactions? 0- it is important for readers to understand

Reviewer #3: (No Response)

**Results**

-Does the analysis presented match the analysis plan?

-Are the results clearly and completely presented?

-Are the figures (Tables, Images) of sufficient quality for clarity?

Reviewer #1: Nothing specific, author have already added the clarifications in results.

Reviewer #2: At mean 40+ mg dose of Prednisolone , how response was ascribed to MTX alone?

Effect on neuritis is not mentioned

Evaluation of inflammatory signs seems subjective

Did adding MTX at 20 mg per week to MDT (dapsone) increased risk of anemia?

Was any attempt made to differentiate from relapses (histopath/slit skin smears?) and drug resistance ?

Reviewer #3: (No Response)

**Conclusions**

-Are the conclusions supported by the data presented?

-Are the limitations of analysis clearly described?

-Do the authors discuss how these data can be helpful to advance our understanding of the topic under study?

-Is public health relevance addressed?

Reviewer #1: Agreed with the improved and re- written conclusions.

Reviewer #2: Need appropriate modification in view of limitations

Reviewer #3: (No Response)

**Editorial and Data Presentation Modifications?**

Reviewer #1: Minor changes in spelling-corrections

Reviewer #2: not clear

Reviewer #3: (No Response)

**Summary and General Comments**

Reviewer #1: Nothing specific.

Reviewer #2: (No Response)

Reviewer #3: (No Response)

PLOS authors have the option to publish the peer review history of their article (what does this mean?). If published, this will include your full peer review and any attached files.

Reviewer #1: **Yes: **Seema Rani

Reviewer #2: No

Reviewer #3: No

---

## [Editor Report · Acceptance letter]

16 Apr 2023

Dear Mme Jaume,

We are delighted to inform you that your manuscript, "Methotrexate as a corticosteroid-sparing agent in leprosy reactions: a French multicenter retrospective study," has been formally accepted for publication in PLOS Neglected Tropical Diseases.

Best regards,

Shaden Kamhawi

co-Editor-in-Chief

Paul Brindley

co-Editor-in-Chief
